# Ancestral African Bats Brought Their Cargo of Pathogenic *Leptospira* to Madagascar under Cover of Colonization Events

**DOI:** 10.3390/pathogens12070859

**Published:** 2023-06-21

**Authors:** Colette Cordonin, Yann Gomard, Ara Monadjem, M. Corrie Schoeman, Gildas Le Minter, Erwan Lagadec, Eduardo S. Gudo, Steven M. Goodman, Koussay Dellagi, Patrick Mavingui, Pablo Tortosa

**Affiliations:** 1Unité Mixte de Recherche PIMIT “Processus Infectieux en Milieu Insulaire Tropical”, Centre National de la Recherche Scientifique 9192, Institut National de la Santé Et de la Recherche Médicale 1187, Institut de Recherche pour le Développement 249, Université de La Réunion, Plateforme de Recherche CYROI, 97490 Sainte Clotilde, Réunionerwan.lagadec69@yahoo.fr (E.L.);; 2Department of Biological Sciences, University of Eswatini, Private Bag 4, Kwaluseni M202, Eswatini; 3Department of Zoology and Entomology, Mammal Research Institute, University of Pretoria, Private Bag 20, Hatfield 0028, South Africa; 4School of Life Sciences, University of KwaZulu-Natal, Durban 4000, South Africa; 5Instituto Nacional de Saúde, Maputo 1008, Mozambique; esamogudojr@gmail.com; 6Negaunee Integrative Research Center, Field Museum of Natural History, Chicago, IL 60605, USA; 7Association Vahatra, BP 3972, Antananarivo 101, Madagascar

**Keywords:** *Leptospira*, bats, Madagascar, continental Africa, evolution, biogeography, structuration

## Abstract

Madagascar is home to an extraordinary diversity of endemic mammals hosting several zoonotic pathogens. Although the African origin of Malagasy mammals has been addressed for a number of volant and terrestrial taxa, the origin of their hosted zoonotic pathogens is currently unknown. Using bats and *Leptospira* infections as a model system, we tested whether Malagasy mammal hosts acquired these infections on the island following colonization events, or alternatively brought these bacteria from continental Africa. We first described the genetic diversity of pathogenic *Leptospira* infecting bats from Mozambique and then tested through analyses of molecular variance (AMOVA) whether the genetic diversity of *Leptospira* hosted by bats from Mozambique, Madagascar and Comoros is structured by geography or by their host phylogeny. This study reveals a wide diversity of *Leptospira* lineages shed by bats from Mozambique. AMOVA strongly supports that the diversity of *Leptospira* sequences obtained from bats sampled in Mozambique, Madagascar, and Comoros is structured according to bat phylogeny. Presented data show that a number of *Leptospira* lineages detected in bat congeners from continental Africa and Madagascar are imbedded within monophyletic clades, strongly suggesting that bat colonists have indeed originally crossed the Mozambique Channel while infected with pathogenic *Leptospira*.

## 1. Introduction

The Malagasy Region, located in the southwestern Indian Ocean (SWIO) and including several islands (Mauritius, Mayotte, La Réunion, Seychelles, and the Union of Comoros), is considered one of the five leading biodiversity hotspots on the planet. These insular territories encompass 0.4% of the earth’s total land surface and host 3.2% of global endemic plant species and 2.8% of global endemic vertebrate species [1]. Such high levels of endemism result from geographic isolation in deep geological time associated with plate tectonics, as well as multiple colonization events (i.e., natural immigration of ancestral populations from source continents, especially Africa) that have led to some of the more pronounced adaptive radiations in the world [2,3,4]. Although Madagascar is home to a remarkable diversity of endemic terrestrial small mammals, the ancestors of all four extant groups (lemurs, tenrecs, rodents, and carnivorans) successfully colonized Madagascar from continental Africa via four asynchronous colonization events during the Cenozoic [5]. The diversity of ecological niches that are typical of the big island further accelerated a series of adaptive radiations leading to the complex modern species composition of Malagasy mammals [2,6]. In addition to terrestrial mammals, Madagascar hosts a notable diversity of bats, with around 46 reported species included in nine different families [7]. A majority of these bat species are endemic to Madagascar and with a few occurring on other islands in the Malagasy Region. On the basis of different phylogenetic inferences, most of this diversity originated from Africa, with the modern Malagasy bat fauna resulting from 28 or 29 independent colonization events [7].

Volant and terrestrial mammals are known reservoirs of a number of viral and bacterial zoonotic pathogens [8,9,10,11,12,13,14]. Among these zoonoses, leptospirosis has been more specifically explored in SWIO wild fauna as it is a disease of major public health concern in the region [15,16,17] besides being amongst the most prevalent bacterial zoonosis worldwide [18,19,20]. Molecular data produced from mammals sampled on SWIO islands have highlighted the biogeography of *Leptospira* in terrestrial and volant mammals from the Malagasy Region. These data have revealed that certain islands host phylogenetically distinct *Leptospira* species/lineages, likely associated with different land mammal adaptive radiations in relatively deep geological time [21]. Altogether, data accumulated through several multidisciplinary research programs have shown that the diversity of pathogenic *Leptospira* in this biodiversity hotspot is mostly shaped by the biogeography of land mammal hosts occurring on each island [21,22].

Nevertheless, the origin of *Leptospira* infections in both volant and terrestrial mammals from this region is currently unknown. Mammals may have acquired these infections in Madagascar following the initial colonization events. Alternatively, colonists may have landed on Madagascar with their cargo of infectious agents. *Leptospira* hosted by Malagasy bats are a relevant model to test these two hypotheses as they are highly diversified while displaying strict host-specificity, and since infection prevalence is high in most bat families [22]. Analyses carried out on Malagasy bats support that the diversity of bat-borne *Leptospira* is best explained by (i) co-diversification of *Leptospira* and their bat hosts and (ii) a few host-switches facilitated by the ecology and behavior of these volant mammals, particularly physical contact in day-roost sites between genetically distinct species or members of different families [22]⁠. Since continental Africa is considered the major source of extant bat taxa in Madagascar [7,23]⁠, we investigated the genetic diversity of *Leptospira* shed by bats in Mozambique, the closest mainland African country to Madagascar, in order to test whether ancestral African bats landed on Madagascar with their pathogenic *Leptospira*, or alternatively acquired these infections following colonization.

## 2. Materials and Methods

### 2.1. Bats Sampling

Two sites were investigated in Mozambique: the Inharosso District and the Gorongosa National Park. A total of 314 bat samples (266 from Inharosso District and 48 from Gorongosa National Park) encompassing seven families were collected between February and July 2015 as part of the Feder Interreg V MOZAR project under the following research permit and ethic approval delivered by the Maputo Museum of Natural History (Ref. 01/MHN/E.27/2015) and the Ministry of Health (N°S/N/SDI/0233/15). Bats were sampled using either mist nets, harp traps, or butterfly nets, as previously described [22]. Tissues were conserved in liquid nitrogen from the field to the lab facilities and then stored in deep freezers until the molecular screening. We follow recent sources for the taxonomy of bats from Madagascar [7] and those from Mozambique [24], except for pipiestrelloids, which are based on a recent revision [25].

### 2.2. DNA Extraction, Leptospira Detection, and Identification

Total nucleic acids were extracted from kidney samples of each collected bat specimen using the Biorobot EZ1 and EZ1 Virus Mini Kit version 2.0 (Qiagen, Hilden, Germany). Samples were spiked with MS2 phage RNA and used as an internal extraction control [26]. A reverse transcription step was performed on total nucleic acids using ProtoScript^®^ II reverse transcriptase (New England Biolabs, Ipswich, MA, USA). Finally, generated cDNAs were used as templates for the detection of pathogenic *Leptospira* by qPCR following a previously described scheme targeting the *rrs* gene [27]. Six negative controls were included on each qPCR assay run on a Stratagene MX3000P thermocycler. Analyses were conducted using the MxPro software, and Ct-values were obtained using default parameters. Identification of pathogenic *Leptospira* at the species level was realized using the *secY* gene, a housekeeping gene [28] commonly used for species identification of pathogenic *Leptospira* infecting small mammals, including bats [21,22]. For each positive sample, a 473-bp-fragment of *secY* gene was amplified by conventional PCR (forward primer: 5′-ATGCCGATCATTTTTGCTTC-3′; reverse primer: 5′-CCGTCCCTTAATTTTAGACTTCTTC-3′), including an initial denaturation step (95 °C, 5 min) followed by 45 cycles of amplification (denaturation at 94 °C for 30 s, annealing at 52 °C for 30 s, extension at 72 °C for 1 min) and a final extension step at 72 °C for 7 min. The amplicons were then sequenced on both strands using Sanger method (Genoscreen, Lille, France).

### 2.3. Phylogenetic Analysis

The consensus sequences of *secY* and sequence alignments were conducted under Geneious software version 9.1.7 [29]. In order to compare the phylogenetic relationships between *Leptospira* from Mozambique and the SWIO region, additional sequences from this region available under GenBank were included in the phylogenies, as well as two sequences from samples collected on Madagascar in the context of another research program (see Appendix A). The best model of sequence evolution was determined using jModelTest2 v2.1.10 [30,31] based on the Akaike Information Criterion (AIC). Phylogenetic constructions were carried out using Bayesian Inference analyses implemented in MrBayes 3.2.6 [32]. The phylogeny constructions consisted of two independent runs of four incrementally heated Metropolis Coupled Markov Chain Monte Carlo (MC3) starting from a random tree. MC3 was run for ten million generations with a subset of trees and associated parameters sampled every 100 generations. The level of convergence was validated with an average standard deviation of split frequencies inferior to 0.05. For each run, the initial 10% of trees were discarded as burn-in, and the consensus phylogeny and posterior probabilities were obtained from the remaining trees.

### 2.4. Structure of Bat-Borne Leptospira

The genetic structure of *Leptospira* diversity was tested through an analysis of molecular variance using the software Arlequin v.3.5.1.3 [33]. In each analysis, a population was defined as all *Leptospira* sequences detected in a bat species at a single sampling site. The geographical structure was tested by grouping populations by sampling site (or country), while host specificity was tested by grouping populations at the level of the host family. The significance levels of the fixation indices were obtained using 1023 non-parametric permutations.

## 3. Results

### 3.1. Positive Samples and Identification

Fifty-nine animals tested positive for pathogenic *Leptospira* infection, representing a global detection rate of 18.8% ± 4.3% (Table 1). At least one specimen was positive for each bat family, with the exception of the Vespertilionidae represented by samples of *Afronycteris nana* and *Scotophilus viridis*. Sequences from a 473-bp-fragment of the *secY* gene were then obtained by conventional PCR from 25 out of the 59 positive specimens from Mozambique. Two additional *secY* sequences were obtained from Madagascar specimens and completed the dataset of this study. The absence of ambiguous peaks allowed the production of good-quality sequences that were deposited in GenBank under accession numbers MT037447—MT037473. Based on the *secY* sequences, a major part of obtained sequences was related to *Leptospira borgpetersenii* (*n* = 13), followed by lineages embedded in clades J (11 samples) and M (1 sample), both with no *Leptospira* assignment (Figure 1). Other previously reported sequences from Mozambique and Madagascar clustered in *L. borgpetersenii*, *Leptospira interrogans*, *Leptospira kirschneri,* and *Leptospira noguchii*.

### 3.2. Phylogeny

The phylogenetic tree depicted in Figure 1 shows that *secY* sequences tend to group according to bat host genera, regardless of the geographic origin of the samples. Indeed, well-supported clades with posterior probabilities ≥ 0.80 are composed of *Leptospira* lineages shed by bat genera common to Mozambique and the Malagasy Region (Figure 1). For instance, clades A and F contain *Leptospira* genotypes found in the family Miniopteridae (genus *Miniopterus*) from Mozambique, Madagascar, and/or Comoros, while clades H, J, and M are composed of *Leptospira* lineages hosted by *Triaenops afer* and *Triaenops menamena* (family Rhinonycteridae), from Mozambique and Madagascar, respectively. Some exceptions to this pattern occur, such as *Nycteris thebaica* (family Nycteridae) and *Paratriaenops furculus* (family Rhinonycteridae), from Mozambique and Madagascar, respectively, and carrying closely related *Leptospira* (clade E), as well as *Scotophilus dinganii* (family Vespertilionidae) and *Miniopterus natalensis* sampled in South Africa and shedding *Leptospira* with genotypes embedded within the same clade (clade C) (Figure 1). The analysis of molecular variance (AMOVA) performed on the *secY* sequences dataset of *Leptospira* in SWIO does not show any geographical structuration of *Leptospira* lineages (*p* = 0.52) but strongly supports a structuration according to the bat host families (*p* < 0.001) (Table 2). Lastly, the topology supports that most bats are infected with *L. borgpetersenii* (clades A–F), in keeping with previous studies [22], while *Leptospira* clustering in the remaining clades requires full MLST genotyping for an assignment of *Leptospira* at the species level.

## 4. Discussion

This study aimed at giving an insight into the relationship between *Leptospira* lineages shed by bats from SWIO islands and, for the most part, those from similar taxonomic groups from south-eastern continental Africa. For this, we described the diversity of pathogenic *Leptospira* hosted by bats from Mozambique and used generated sequence data to construct a phylogeny comprising samples from SWIO islands and continental Africa using the polymorphic *secY* gene marker, which has been shown to be highly discriminatory [21,34]. We performed an analysis of molecular variance to test whether this genetic diversity is structured by geography or by the phylogeny of their hosts.

We obtained *secY* sequences for 25 of the 59 samples testing positive through qPCR, which falls within the success rate obtained for comparative studies in the region [22,35], and which likely results from the higher sensitivity of qPCR as compared to end-point *secY* PCR used prior sequencing. Of note, no *secY* sequence could be obtained for any *Leptospira*-positive sample with Ct-value > 35 (N = 24), strongly suggesting that the success of *secY* PCR/sequencing is conditioned by the *Leptospira* load in each sample. In addition, among the 35 *Leptospira*-positive samples with Ct-values < 35, 10 samples did not allow PCR amplification with *secY* conventional primers, which can be best explained by the fact that bat-borne *Leptospira* is quite distinct from a rodent- or cattle-borne *Leptospira*, which genomes have been used to develop MLST primers. To our knowledge, there is no available full genome for any *Leptospira* shed by bats, which is urgently needed to design new primers and molecular tools required for further exploring the diversity of these bacteria.

Our results reveal a wide diversity of *Leptospira* lineages shed by bats from Mozambique that are mainly related to *L. borgpetersenii*, *L. interrogans*, *L. kirschneri,* and *L. noguchii*. We cannot rule out the possibility of multiple infections in bats, as previously reported for small terrestrial mammals [36], since the screening method used was not designed for detecting mixed infections. However, no multiple peaks were detected in sequencing chromatograms, suggesting that multiple infections are, at most, a minority. Importantly, molecular variance analysis supports that the diversity of *Leptospira* sequences obtained from bats sampled in Mozambique, Madagascar, and Comoros is structured according to their phylogenetic origin (at the family level) rather than to geography related to the sympatric occurrence of different bat families within the local community. Such structuration is similar to that previously reported on Madagascar, where each bat family sheds one or a handful of *Leptospira* lineages [21,22,35].

Our results support that certain lineages of ancestral bats colonizing Madagascar from eastern Africa were actually infected by *Leptospira* before arriving on the island. Indeed, *Leptospira* lineages shed by African and Malagasy Miniopteridae are embedded in the same monophyletic clade (clade A, Figure 1), indicating a common origin of these lineages. The same pattern is observed with *Leptospira* shed by bats of the family Rhinonycteridae from Mozambique and Madagascar (clade J, Figure 1). By contrast, some clades of *Leptospira* contain lineages shared by different families: clade C contains lineages shed by Miniopteridae and Vespertilionidae, while clade E contains *Leptospira* lineages shed by Nycteridae and Hipposideridae. Such patterns contradicting the host specificity of *Leptospira* are indicative of host switches that could have occurred among African bats, as previously suggested for *Leptospira* hosted by *Miniopterus* spp. and *Myotis goudoti* on Madagascar [22]. It is now well-documented that the origin of the extant land mammals of Madagascar can be explained by four successful colonization events and subsequent large-scale speciation; each of the ancestral groups appears to be of African origin and reached the island via water dispersal on some form of floating vegetation [6,37]. However, given their flight capacity and relative ease in crossing the Mozambique Channel, bats at the family level have colonized Madagascar, probably from the eastern portions of Africa, on about 28 independent occasions [7]. In this context, complementary screening work needs to be conducted on the *Leptospira* of eastern African bats to provide a greater context of the relationship between different Afro-Malagasy forms of these bacteria, as well as their biogeography. Altogether, the natural history of Malagasy mammals and their associated microbial biomes suggest that the current diversity of pathogenic *Leptospira* on the island results from colonization of mammalian reservoirs from continental Africa and subsequent co-diversification processes.

It is noteworthy that a study conducted on coronaviruses in bats from the same regions has also highlighted host specificity between these viruses and their host with low levels of host switches [12]. Therefore, data produced using bacteria and viruses infecting bats from Mozambique, Madagascar, and the Indian Ocean islands support that the geographic isolation of Madagascar has promoted the radiation and strong adaptation of endemic pathogens to their specific hosts, with some levels of host-switch, which at least in part has to do with local bat community ecology, such as non-related bats using the same day roost sites.

## Figures and Tables

**Figure 1 pathogens-12-00859-f001:**
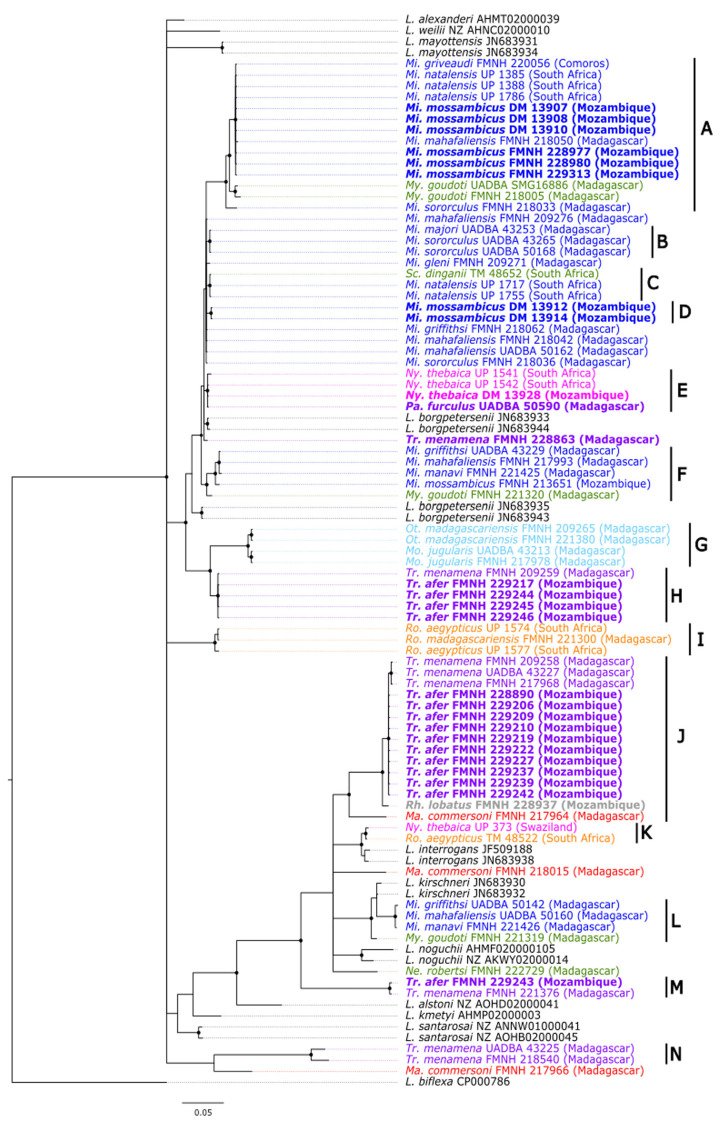
*Leptospira* phylogenetic tree based on a 473-bp *secY* fragment from bat samples from the Malagasy region and neighboring Africa. Bayesian inference with HKY + I + G substitution model was used for analysis. Nodes with a black dot correspond to posterior probabilities ≥ 0.80. *Leptospira* haplotypes are colored according to host family (blue: Miniopteridae, green: Vespertilionidae, pink: Nycteridae, red: Hipposideridae, purple: Rhinonycteridae, light blue: Molossidae, orange: Pteropodidae). Sequence names in bold indicate sequences obtained during the course of this study and include 25 sequences from Mozambique together with two sequences from Madagascar. Mi: *Miniopterus*, My: *Myotis*, Sc: *Scotophilus*, Ny: *Nycteris*, Pa: *Paratriaenops*, Tr: *Triaenops*, Ot: *Otomops*, Mo: *Mormopterus*, Ro: *Rousettus*, Ma: *Macronycteris*. The letters A–N refer to monophyletic clades of *Leptospira* sequences obtained from bats in the southwestern Indian ocean region.

**Table 1 pathogens-12-00859-t001:** Results of *Leptospira* detection by qPCR for bat species from Mozambique.

Bat Family	Bat Species	Sample Size	Number of Positives (% ± 95% CI)	Number of *secY* Positive Samples	*Leptospira* Clade	*Leptospira* Species
Hipposideridae (*n* = 65)	*Hipposideros caffer*	65	2 (3.1% ± 4.2%)	0	-	-
Miniopteridae (*n* = 31)	*Miniopterus mossambicus*	31	15 (48.4% ± 17.6%)	8	A, D	*L. borgpetersenii*
Molossidae (*n* = 67)	*Chaerephon pumilus*	10	0 (0.0% ± 0.0%)	0	-	-
	*Mops condylurus*	57	3 (5.3% ± 5.8%)	0	-	-
Nycteridae (*n* = 19)	*Nycteris thebaica*	19	4 (21.1% ± 18.3%)	1	E	Unknown
Rhinolophidae (*n* = 73)	*Rhinolophus* cf. *swinnyi*	2	0 (0.0% ± 0.0%)	0	-	-
	*Rhinolophus landeri*	10	3 (30.0% ± 28.4%)	0	-	-
	*Rhinolophus lobatus*	8	2 (25.0% ± 30.0%)	1	J	Unknown
	*Rhinolophus mossambicus*	19	0 (0.0% ± 0.0%)	0	-	-
	*Rhinolophus rhodesiae*	33	2 (6.1% ± 8.1%)	0	-	-
	*Rhinolophus* sp.	1	0 (0.0% ± 0.0%)	0	-	-
Rhinonycteridae (*n* = 51)	*Triaenops afer*	51	28 (54.9% ± 13.7%)	15	H, J, M	Unknown
Vespertilionidae (*n* = 8)	*Afronycteris nana*	6	0 (0.0% ± 0.0%)	0	-	-
	*Scotophilus viridis*	2	0 (0.0% ± 0.0%)	0	-	-
Total		314	59 (18.8% ± 4.3%)	25		

**Table 2 pathogens-12-00859-t002:** Analysis of molecular variance (AMOVA) based on *Leptospira secY* gene. In each analysis, a population is defined as all *Leptospira* sequences detected in a single bat species at a single sampling site.

Comparison	Source of Variation	df	Fixation Indices (F)	*p*-Value	% Variation
Sampling sites	Among sites (*n* = 5)	4	FCT = −0.00275	0.51992	−0.28
	Among host species within sites	19	FSC = 0.20315	0.00285	20.37
	Within host species	32	FST = 0.02096	0.00285	79.90
Host families	Among families	7	FCT = 0.28539	<0.001	28.64
	Among host species within families	16	FSC = 0.05958	0.041082	−4.26
	Within host species	32	FST = 0.24282	0.00569	75.72

## Data Availability

Information is available in the Appendix A.

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
