# Peer review of "Ancestral African Bats Brought Their Cargo of Pathogenic Leptospira to Madagascar under Cover of Colonization Events"

_pathogens, 2023, doi:10.3390/pathogens12070859_

Round 1

Reviewer 1 Report

In this work, authors aimed to ensure whether Malagasy mammal hosts acquired leptospiral infections on the island following colonization events, or alternatively brought leptospires from continental Africa. Through analyses of molecular variance of Leptospira, they suggested that bat colonists have indeed originally crossed the Mozambique Channel while infected with pathogenic Leptospira. This paper provides new insights into the origin of pathogenic leptospira hosted by bats in Madagascar, but some information is missing which reduces the reliability of the conclusions of the paper.

Major comments:

1. The spell of “Leptospria” should be uniform, such as line 97, 128, and 224.

2. In the methods section, two sites were investigated in Mozambique, authors should clearly describe how many samples were collected in the two sites, respectively. And the sampling time of these samples was also missing. It's better to draw a map of the sampling points.

3. The primers used in this study should be written and the PCR reaction procedure should also be described in detail.

4. Negative controls should be set for qPCR to clarify CT thresholds.

5. Whether the diversity of Leptospira in environmental samples from the sampling area is consistent with that carried by bats.

6. In the Discussion section, authors should further discuss the methods used to determine pathogen sources in different regions and the areas that need to be optimized to enhance the conclusions of the study in this paper.

The quality of English language is high, reading this paper is enjoyable.

Author Response

Please note that each comment is italicized in black while our answer to each specific comment is typed in blue. The line numbering corresponds to the R1 version of the ms without track changes. 

  1. The spell of “Leptospria” should be uniform, such as line 97, 128, and 224.

We italicized Leptospira throughout the ms, as requested. Please note that Leptospira was not italicized in the subtitles of the originally submitted ms because these were already italicized.

  1. In the methods section, two sites were investigated in Mozambique, authors should clearly describe how many samples were collected in the two sites, respectively. And the sampling time of these samples was also missing. It's better to draw a map of the sampling points.

Lines 89-91: details on the sampling sites and sampling dates were added to the revised ms

  1. The primers used in this study should be written and the PCR reaction procedure should also be described in detail.

This information is now included in the revised ms (see lines 113-118)

  1. Negative controls should be set for qPCR to clarify CT thresholds.

This information is now included in the revised ms (see lines 107-110)

  1. Whether the diversity of Leptospira in environmental samples from the sampling area is consistent with that carried by bats.

No environmental samples were analyzed in this investigation

  1. In the Discussion section, authors should further discuss the methods used to determine pathogen sources in different regions and the areas that need to be optimized to enhance the conclusions of the study in this paper.

We are grateful to the reviewer for this point.  We have added some new text to the manuscript regarding the origin of Malagasy mammals. We have also added a sentence for additional work on bat-borne Leptospira from eastern Africa (see lines 242-251)

Reviewer 2 Report

This is a well written article. I just have a few comments:

General - It would have been interesting to see if phylogenies of other genes reflected that seen for the SecY gene. Was a multi-locus approach considered?

Methods - A brief description of bat taxonomy would be helpful for people less familiar with the subject.

Lines 114-155 state 'additional sequences from this region available under GenBank were included in the phylogenies'. Were all available sequences included or what was the criteria for including sequences?

Lines 136-138 need to be removed as these seem to have been left over from the template.

Author Response

Please note that each comment is italicized in black while our answer to each specific comment is typed in blue. The line numbering corresponds to the R1 version of the ms without track changes.

1. General - It would have been interesting to see if phylogenies of other genes reflected that seen for the SecY gene. Was a multi-locus approach considered?

As mentioned in the ms (lines 110-112), secY is the most resolutive marker within the MLST scheme3 that is commonly used in the western Indian Ocean region. In addition, full MLST is more challenging on wild fauna samples than on clinical samples, which display high bacteremia. Therefore full MLST would have dramatically reduced the number of genotyped samples and hence preclude robust interpretations.

2. Methods - A brief description of bat taxonomy would be helpful for people less familiar with the subject.

Some additional details have been added to the text associated with the reviewer's comment on the taxonomic sources used in the paper associated with bat scientific names (see lines 96-98).

3. Lines 114-155 state 'additional sequences from this region available under GenBank were included in the phylogenies'. Were all available sequences included or what was the criteria for including sequences?

We selected sequences obtained from a large panel of bat species and representing each known Leptospira clade from different locations in southwestern Indian Ocean.

4. Lines 136-138 need to be removed as these seem to have been left over from the template.

We apologize for this. This section has been removed.

Reviewer 3 Report

L137-139. this sentence should be removed as it is an indication from the journal's template

Figure 1 is difficult to interpret, specially what part of the graph corresponds to leptospira classification and what to bats species. Please consider the possibility of 2 figures (one for each?)

Author Response

Please note that each comment is italicized in black while our answer to each specific comment is typed in blue. The line numbering corresponds to the R1 version of the ms without track changes.

1. L137-139. this sentence should be removed as it is an indication from the journal's template

We apologize for this. This section has been removed.

2. Figure 1 is difficult to interpret, specially what part of the graph corresponds to leptospira classification and what to bats species. Please consider the possibility of 2 figures (one for each?)

It is not possible to split this figure in 2 since it represents the phylogeny of Leptospira with branches colored according to host family. We have slightly modified the legend, which now reads « Leptospira phylogenetic tree based on a 473-bp secY fragment… » (line 182).

Round 2

Reviewer 1 Report

  • All the problems have been dealt with.